# A Proposed Methodology to Evaluate Machine Learning Models at Near-Upper-Bound Predictive Performance—Some Practical Cases from the Steel Industry

Leo S. Carlsson and Peter B. Samuelsson *

Royal Institute of Technology, Brinellvägen 23, 114 28 Stockholm, Sweden; leoc@kth.se
* Correspondence: petersam@kth.se

**Abstract:** The present work aims to answer three essential research questions (RQs) that have previously not been explicitly dealt with in the field of applied machine learning (ML) in steel process engineering. RQ1: How many training data points are needed to create a model with near-upper-bound predictive performance on test data? RQ2: What is the near-upper-bound predictive performance on test data? RQ3: For how long can a model be used before its predictive performance starts to decrease? A methodology to answer these RQs is proposed. The methodology uses a developed sampling algorithm that samples numerous unique training and test datasets. Each sample was used to create one ML model. The predictive performance of the resulting ML models was analyzed using common statistical tools. The proposed methodology was applied to four disparate datasets from the steel industry in order to externally validate the experimental results. It was shown that the proposed methodology can be used to answer each of the three RQs. Furthermore, a few findings that contradict established ML knowledge were also found during the application of the proposed methodology.

**Keywords:** machine learning; stability; predictive performance; statistical modeling; electric arc furnace; vacuum tank degasser; ladle refining furnace; secondary metallurgy

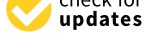



## 1. Introduction

In light of the plethora of challenges facing the steel industry, machine learning (ML) is seen as a promising technology that can overcome many of these challenges. As opposed to deterministic models, ML models do not require explicitly defined relations between the variables used to create the model. This is advantageous in steel processes where some of the variables are difficult to accurately quantify or where the relationship between certain ongoing variables and the output variable cannot be analytically defined. As such, there have been numerous studies that have used ML models, which are in fact statistical models, to resolve challenges in a wide range of processes and optimization problems within the steel industry. Predicting the electrical energy (EE) consumption of the electric arc furnace (EAF) [1], the tap temperature of the EAF [2], the temperature of molten steel during treatment in secondary metallurgy [3], end-point prediction of temperature and alloying elements in the basic oxygen furnace (BOF) [4], and prediction of the molten steel temperature in the steel ladle and tundish [5] are several examples of ML models applied in the context of steel process engineering.

Despite the prominent research activity in the field of applied ML in steel process engineering, there are many practical challenges that have to be solved before ML can be substantially leveraged for value creation within the steel industry. These challenges are not specific to the steel industry itself, rather they are challenges that any industry faces when attempting to successfully utilize ML models. Some of these challenges are specific to the implementation, supervision, and maintenance of ML models while other challenges are related to the infrastructure and work methodology governing both the data

and the ML model systems [6–9]. The data governing the training and testing of any ML model set the upper-bound predictive performance and the trustworthiness in the eyes of domain experts for that ML model. *In this work, the upper-bound predictive performance is defined as the highest predictive performance that can be achieved by a stable ML model on a finite test dataset*. The goal of the present work is to answer three research questions (RQs) that have previously not been explicitly dealt with in the field of applied ML in steel process engineering. These essential RQs are:

1. How many training data points are needed to create a model with near-upper-bound predictive performance on test data?
2. What is the near-upper-bound predictive performance on test data?
3. For how long can a model be used before its predictive performance starts to decrease?

By training and evaluating ML models on numerous samples of training data and test data, it could be possible to answer the three RQs. The answers can then be used by both practitioners and researchers in the field to better understand the potential of using ML models for an arbitrary steel process under study. For example, expectations of the predictive performance of ML models for a specific process can be established before investing resources in implementing the model in production. These resources could instead be used to improve the quality of the data governing the performance of the ML model. After all, the data quality always sets the upper-bound predictive performance of any ML model and is one of the main challenges facing the steel industry in regards to the practical usability of ML models [10–12].

The present work is unique for three reasons. First, previous research in applied ML in steel process engineering has not evaluated the amount of training data required to create an ML model that achieves a near-upper-bound predictive performance on test data for the steel processes under study. The generally accepted practice is to use as many data points as possible as training data. However, due to changes that occur in steel processes over time, this recommendation is not sound because the large number of historical samples could be irrelevant to the current state of the process. Reasonably, the most recent samples are more relevant to use as training samples due to the wear of equipment and change in production practices over time, all of which impose changes in the distributions governing the data. In addition, the number of training samples for any steel process is proportional to the number of produced heats. The number of produced heats per year in steel plants is commonly within the range $10^3$ to $10^4$. By having an understanding of the number of required training samples, it is possible to assess when it is possible to have a model that achieves a satisfactory predictive performance after, for example, a maintenance overhaul of the process equipment. This type of equipment intervention could affect the process data to such an extent that the predictive performance of the model is decreased. Second, the current available research in the field of steel process engineering has not investigated the amount of test data, in chronological order after the training data, that a model can predict before the predictive performance of the model decreases. All processes in the steel industry change due to, for example, wear of the equipment and changes in standard operating procedures (SOP). Naturally, these changes will be reflected in the data collected from the processes. Hence, it is expected that the model performance on previously unseen data, i.e., test data, will deteriorate the longer the model is used after training. Lastly, to externally validate the experimental results, four datasets encompassing four processes from three different steel plants will be used. Each of these datasets represents produced heats over 2 to 14 years of continuous steel production.

## 2. Background

### 2.1. Upper-Bound Predictive Performance

The highest possible predictive performance of any ML model is limited by how well the data used to train the ML model represent reality and how well the ML model can adapt its weights to accurately capture the relationship between the input variable and the output variable. Since the ML field has mainly focused on developing ML model algorithms that

can adapt the resulting model to complex data [12,13], the most common limitation to the predictive performance of ML models is how well the data represent reality. For the steel industry, the datasets produced by the processes are limited to the number of produced heats per year, which is usually within the range of $10^3$ to $10^4$. Furthermore, the number of variables needed to predict various metallurgical phenomena in steel process engineering are limited by physico-chemical relations. This further eliminates the need for complex ML models since the number of relevant input variables is typically within the range of $10^1$ to $10^2$.

In the current paper, the following definition is used: *the upper-bound predictive performance is the highest predictive performance that can be achieved by a specific ML model algorithm on a finite test dataset that follows in chronological order from the training dataset used to adapt the parameters of the resulting model. The model producing the upper-bound predictive performance must be stable, i.e., the predictive performance must not be the result of a randomization procedure within the ML model algorithm.*

Near-upper-bound predictive performance builds on the aforementioned definition of upper-bound predictive performance. The term *near* illuminates the fact that the upper-bound predictive performance is the highest predictive performance that can be achieved. How close a model comes to this theoretical limit is partly a random outcome since the ML model algorithm used in this paper utilizes randomization procedures during the model training phase. Near-upper-bound predictive performance should therefore be interpreted as the highest predictive performance that can be achieved, per the above definition, given the randomization imposed by the ML model algorithm.

### 2.2. Applied Machine Learning in Steel Processes

The advantage of using ML models is that they do not require explicitly defined relationships between the input variables and the output variable. Therefore, ML models can be used to predict physical phenomena in steel processes that are dependent on factors that are difficult to accurately quantify. The present work demarcates the application of ML models in the steel industry to two prediction problems that are known to be difficult to model using deterministic models such as mass- and energy balance equations. These are the predictions of the EE consumption in the EAF and the end-point temperature in secondary metallurgy. Examples illuminating the reasons why these two problems are difficult to model using deterministic models will be provided in the following sections along with the description of each of the processes governing the four datasets used in the present work.

#### 2.2.1. Electrical Energy Consumption of the Electric Arc Furnace

The EAF is the main melting process in mini-mill types of steel plants. It commonly uses steel scrap from various recycling operators, alloys, and direct reduced iron (DRI) as raw materials to produce molten steel. The molten steel is then further processed in downstream processes of the steel plant.

A general EAF process begins with the charging of raw materials into the furnace using a scrap bucket. The electrodes of the EAF are then ignited and bored down into the raw material This phase, commonly known as the melting phase, continues until enough scrap has melted to make room for a second bucket of raw materials. After the second charge of raw materials has been added, the process continues in yet another melting phase. A third bucket of raw materials may be added if the target weight of steel has not been met. Some steel plants also use burners to remove the cold spots that appear in the furnace wall furthest away from the electric arc during the melting phase. After the majority of the raw material has melted, the refining phase starts during which the furnace operator has the ability to adjust the steel to its pre-specified composition. Carbon (and silicon) can also be added together with oxygen lancing to facilitate heating through exothermic chemical reactions. Any type of chemical heating reduces the amount of EE needed to either melt the raw materials or raise the temperature of the molten steel. When the molten steel has reached its target temperature and composition, it is tapped into a ladle for further



downstream processing. Before the start of the next heat, any required preparations are made such as the fettling of the refractories.

One of the main costs of operating the EAF is electricity. Hence, a decrease in EE consumption will result in a reduction in costs when producing steel in mini-mills. A practical application of an ML model, predicting the EE consumption, is that the predicted EE consumption can be used by the EAF process control system operator as a reference to optimize the process towards lower EE consumption. Quantifying the effect of various scrap types on EE consumption is one of the main challenges in accurately predicting EE consumption using deterministic models. The melting of scrap in the EAF depends partly on the volume-to-surface area ratio for the charged types of scraps with various shapes and apparent densities. Furthermore, the layering of the different types of scraps in the scrap bucket, and thus also in the furnace, affects the melting behaviour of the scrap during the melting phase. While there exists deterministic equations for the melting of individual scrap pieces [14], equations that accurately quantify the effect of arbitrary blends and volumes of various types of scraps with different shapes and apparent densities do not exist. The use of ML modeling is therefore a valid choice of modeling framework for predicting EAF's EE consumption.

The present work will use data from two EAFs, which will be referred to as EAF1 and EAF2, respectively. EAF1 produces stainless steel and EAF2 produces steel for ball bearing rings, rods, and engineering steels. Both EAFs uses only scrap and alloys as raw materials.

### 2.2.2. End-Point Temperature in Secondary Metallurgy

During secondary metallurgy, the steel melt composition is further adjusted to match the composition of the specific steel grade that is to be produced. In the present work, the Ladle Refining Furnace (LRF) and Vacuum Tank Degasser (VTD) processes are of interest.

In the LRF, the target composition of the steel grade to be produced is achieved by both removing unwanted elements and by adding wanted elements. Unwanted elements, such as oxygen, are removed by adding deoxidation agents such as aluminum and ferrosilicon while stirring the steel melt. Stirring is facilitated by injecting argon through porous plugs in the bottom of the ladle as well as by injecting argon from the top of the melt using a lance. The addition of wanted elements depends on the type of steel grade to be produced. Common alloys are manganese and molybdenum. If the steel melt temperature is below the target end temperature, the steel is heated using electricity prior to the transportation to the continuous casting machine (CCM).

The purpose of the VTD is to reduce the level of hydrogen and nitrogen present in the steel melt. This is conducted by inserting the ladle into the vacuum chamber where the pressure is decreased to an extent that facilitates the removal of dissolved nitrogen and hydrogen to their gaseous equivalents. To hasten this process, argon can be injected during the vacuum treatment process through porous plugs located in the bottom of the ladle. The tolerance levels of the dissolved gases vary by steel grade and the time under vacuum pressure will therefore vary. Naturally, a longer vacuum treatment leads to a lower temperature of the steel melt which has to be compensated for by external heating.

The importance of achieving the specific target temperature at the end of the LRF and VTD processes is due to the pre-specified arrival temperature at the CCM. If the temperature of the molten steel arriving at the CCM is too high, the molten steel will be more difficult to solidify in the oscillating mold. If the steel does not solidify in the mold, a breakthrough will happen where molten steel pours out of the mold and damages adjacent parts of the CCM. The severe costs of a breakout are not only associated with repairs but also with the downtime in the steel plant that is inevitable during a subsequent reparation campaign. If the arrival temperature is too low, then the ladle must be sent back for reheating, which requires re-planning of the present ladle and the following ladles whose path gets interrupted by the re-planning of the present ladle. Inevitably, this incurs a loss in productivity and therefore additional costs for the steel plant.

The challenge in predicting the end-point temperature of any secondary metallurgy process is partly related to the ladle heat loss. The heat loss in the ladle is substantially dependent on the thickness of the ladle walls, which is reduced the longer the ladle is used to transport molten steel. The ladle wear rate depends not only on its total time of use, but also on the type of steel produced due to the different effects various steel compositions have on the ladle refractory walls. Accurately quantifying the total ladle wear and its effect on heat loss is therefore difficult using deterministic models. This is one of the reasons why ML models are a valid choice in predicting the end-point temperature of processes in secondary metallurgy.

The VTD process in the present work resides within the same steel plant as the LRF process described in the previous section. In addition, the VTD and LRF processes reside within the same unit called the secondary metallurgy station. This means that, after the VTD process is finished, the operator has the ability to move the ladle to the LRF section of the secondary metallurgy station for additional treatment. This is often carried out to adjust the temperature of the molten steel such that it arrives at the CCM at a pre-specified arrival temperature. The two processes will be referred to as LRF and VTD, respectively, in the following sections of this paper.

Figure 1 illustrates the residence of each of the four processes within the three steel plants providing the data used in the present study.

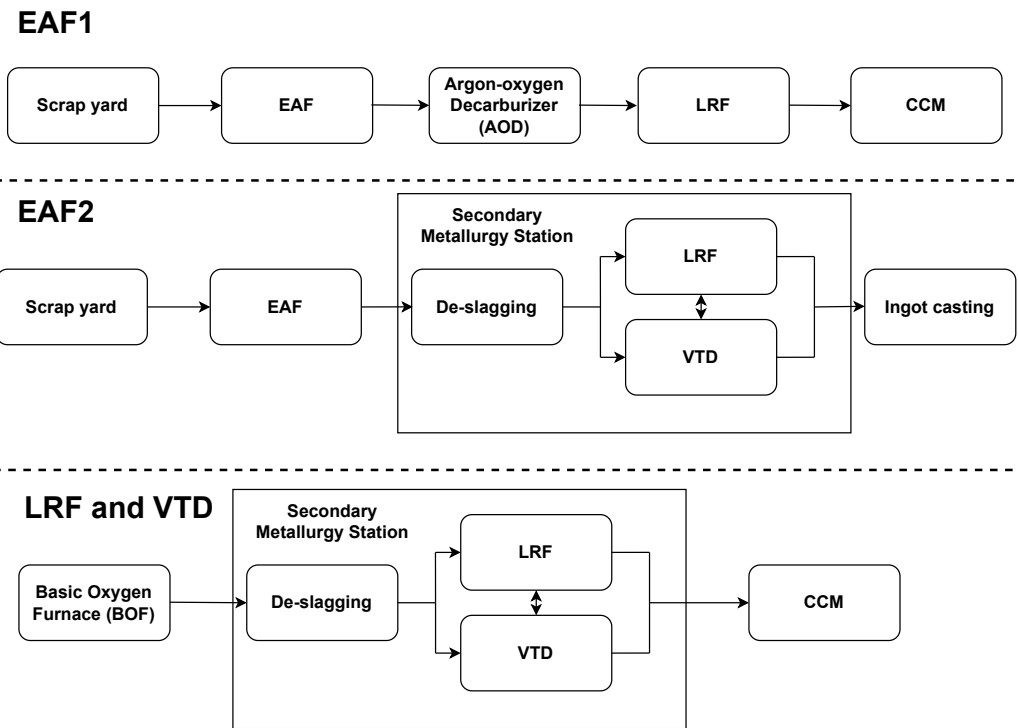

**Figure 1.** The location of each process EAF1, EAF2, LRF, and VTD, with respect to the upstream and downstream processes in each steel plant.

2.2.3. Previous Work

There is a plethora of published research on using ML models to predict the EE consumption of the EAF process and the end-point temperatures of the LRF and VTD processes. A review of publications creating or using ML models to predict the EE consumption of the EAF was published in 2019 [1], which included models starting from the early 1980s. The review compiled information on variables used to create the models, the predictive performance of each model, and the type of EAFs governing the data, with recommendations for future work based on identified shortcomings. These recommendations were adhered to in some further developments [15,16]. Additional developments include the comparison

of the predictive performances from three ML trained on data from five different EAF [17] and the evaluation of multiple ML models predicting the EE consumption with the aim to retrieve the optimal melting time for a given heat [18]. The published papers on ML models predicting the LRF end-point temperature are numerous [19–27]. However, the research primarily concerns the comparison of the predictive performance of models created using existing or newly developed ML modeling frameworks. Some notable examples are the use of a wavelet transform-based weighting algorithm for the support vector machine (SVM) framework [21] and the use of an outlier detection component to either replace or remove outliers in the dataset [23]. The published ML models predicting the end-point temperature of the VTD process are more sparse but follow the same regimen as the corresponding research on the LRF process [28–30].

The focus of the cited previous work has mainly been to create a model with as high predictive performance as possible using various ML modeling frameworks. While a model with high predictive performance is important, the overall focus on this aspect of ML models has left the field with several unconsidered aspects that are of practical importance. One of these aspects is related to the number of training data points that are needed to create an ML model with a near-upper-bound predictive performance. The most commonly used approach in the field of applied ML in steel processes is to use as many training data points as possible. Commonly, the total number of data points is divided into 80/20 between the training and test data. However, the number of required training data points could possibly be lower since older data are less relevant than new data for a model whose aim is to make predictions using data from future heats. This is related to the changes that continuously occur in production that will inevitably be reflected in the data. Changes that occur could be, for example, changes in SOP and varying demand for certain steel grades produced by the steel plant. Another aspect is related to how long the model can be used before its predictive performance starts to deteriorate. This could be evaluated by calculating the predictive performance on test datasets with varying numbers of test data points. The near-upper-bound predictive performance on test data is another unexplored aspect. This could also be evaluated by calculating the predictive performance on test data for a varying number of test data points. These aspects are the basis of the RQs of the present work which will be formulated in Section 2.4. The importance of answering these questions for both practitioners and researchers in the field will also be illuminated.

### 2.3. Data from Steel Processes

All ML models, which in fact are statistical models, rely on data both to adapt their weights during the training phase and to predict previously unseen samples during the testing and deployment phases. Consequently, the data will always limit the upper-bound predictive performance of any ML model and, therefore, also determine its practical usefulness. Data originating from steel processes are, by experience, of varying quality and subject to frequent change.

#### 2.3.1. Data Quality

Data quality refers to how well the data represent reality. For example, temperature measurement of molten steel using a temperature probe or the weight of ferro-silicon registered by a scale prior to being added into the LRF. While the goal of any measurement is to attain a value that is as close as possible to the true value, there are numerous sources that limit the possibility of achieving this goal. Examples of sources of errors in data originating from steel processes are measurement limitations, equipment limitations, inconsistent data collection and transformation procedures. An example of equipment limitations is the precision of the temperature probe used to measure the temperature of molten steel. While the error is typically low, it will still limit any ML model to predict the temperature of molten steel with a higher precision than the precision of the temperature probe. Building on the aforementioned example, limitations in accurately measuring the temperature of molten steel are particularly prevalent in the EAF process. For example,

the presence of solid scrap in the steel melt where the measurement is taken will produce a value that underestimates the temperature of the steel melt. The opposite is true if the temperature probe registers the temperature in the vicinity of the electrodes. In both of these cases, an additional temperature sample is often taken. However, the erroneous values must be either flagged or removed from the process database in order to remove the risk of using these values when developing an ML model.

### 2.3.2. Distribution Change

The distribution of values from variables originating from the steel process is subject to continuous change due to the changes in the processes that occur over time. If the distribution of the variables governing an ML model changes from the training data to the test data, then it is expected that the predictive performance on test data will be worse than on the training data. This is because the weights of an ML model are adapted to the training data and then used to predict data previously unseen by the model from the model testing and model deployment phases. For example, the scrap yard in a mini-mill steel plant consists of many different scrap types. These scrap types are combined in various amounts in scrap buckets using pre-specified recipes. The recipes commonly provide some flexibility by allowing the use of alternative scrap types. This makes it possible to continue producing certain steel grades even though a specific scrap type may be unavailable. During these circumstances, the distribution of the variables reflecting the charged weight of the scrap types will change. The frequency of this type of change is dependent on market-based factors such as demand for certain steel grades as well as the supply of scrap types from scrap vendors. It is likely that these exemplified distribution changes will impact the predictive performance of an ML model predicting the EE consumption of an EAF. It is well known that the average size and shape of a scrap type affects the EE consumption during the duration of the melting phase of the EAF process. In addition, if the change in charged scrap types is frequent, one could interpret these input variables as noisy. Noisier data makes it more difficult for any model to predict well. As a contrasting example, the molten steel transported to the LRF and VTD processes in an integrated steel mill has a less profound change in its distribution with respect to the weight and temperature of the molten steel. Of course, should the SOP of the LRF or VTD process change, then the distribution of the temperature could also change.

### 2.3.3. Traceability

The ability to trace variability in data quality and changes in the distribution of data used to create ML models is important. Based on experience, however, steel plants generally lack systems that detect whenever there is a data quality issue or a significant change in the distribution of the data. This makes it difficult for those that are responsible for the ML model performance to be proactive in retraining the model on data that reflects recent changes.

The limitations in traceability also makes it difficult to know which historical data is relevant for training an ML model. After all, the goal of any ML model is to predict well on new data points. Newer data points will be different from older data points due to the changes in production that occur continuously due to, for example, seasonal effects and improvements in SOP. As mentioned earlier, it is common to use as much training data as possible, usually by dividing the total number of data points into a training set and test set using an 80/20 split. However, this is an arbitrary approach that has its roots in general ML model development and is not based on a sophisticated analysis of distribution changes, let alone considerations rooted in the application domain. Major and sudden changes in distributions can most likely be pin-pointed to larger maintenance overhauls and changes in SOP. However, changes that occur slowly over time are more difficult to detect. In addition, there could be multiple confounding factors that influence the change in a distribution. For example, the EE consumption of the EAF is influenced by factors such as scrap types used, the amount of scrap, and the tap-to-tap time. To conduct a comprehensive

analysis, the distributions of all variables that significantly affect the EE consumption must be analyzed. Furthermore, the effect of one variable on the EE could cancel out the effect of a second variable. For example, if the tap-to-tap time decreases while the amount of charged scrap increases, the EE consumption could theoretically stay the same.

While it is possible to conduct a distribution analysis for all variables used for training an ML model, the purpose of the present work is to develop a proposed methodology that illuminates the consequences of the distribution changes on the predictive performance of ML models. The purpose of the present work is not to analyze the reasons as to why the distributions change. The authors, however, acknowledge that this kind of analysis is an interesting direction for future work.

*2.4. Research Questions*

As described in Section 2.2.3, previous research in applied ML in steel process engineering has left out several considerations that are of practical importance. Therefore, the present work aims to propose a methodology to fellow researchers and practitioners in the steel industry that answers the following RQs:

**RQ1:** How many training data points are needed to create a model with near-upper-bound predictive performance on test data? An answer to this question will make it possible for a practitioner to determine the number of heats that has to be produced before it is possible to create a model whose usefulness in operative decision making can be determined.

**RQ2:** What is the near-upper-bound predictive performance on test data? By being able to answer this question, a practitioner can gauge whether the predictive performance is high enough for the model to be useful in practice or if effort instead should be put on improving, for example, the quality of the data.

**RQ3:** For how long can a model be used before its predictive performance starts to decrease? With an answer to this question, a practitioner can better understand when the predictive performance of the model is expected to deteriorate. Subsequently, the practitioner can then establish model governance that determines how often the model needs to be retrained in order to continuously uphold its predictive performance over the long-term.

From the perspective of the researchers, the answers to the three questions will provide important information when comparing ML models with other modeling approaches such as physico-chemical, i.e., deterministic, models of a steel process of interest.

## 3. Method

The methodology governing the experiments can be divided into Model Construction (Section 3.2) and Unbiased Evaluation (Section 3.3).

The Model Construction methodology aims to find the most optimal model type with respect to predictive performance, stability, and complexity. The predictive performance of any model type is important since it quantifies the accuracy of the model type, i.e., how well it predicts the output variable on previously unseen data. A model type, in the present work, is defined as a model with a specific set of input variables, output variable, parameters, dataset used to train the model and dataset used to test the model. Since the Artificial Neural Network (ANN) algorithm, which is the ML model algorithm used in the present work, use various randomization procedures, the result of instances of these model types will, to some extent, be a random outcome. A stable model type achieves similar predictive performance regardless of instance, given that its input variables and parameters are the same. Lastly, the complexity of the model type should be kept as low as possible by adhering to the principle of parsimony. This principle states that the simplest solution should be selected if there are two or more equally well performing solutions. Each step of the Model Construction will be briefly explained in Section 3.2. See previous work for a detailed explanation of the Model Construction methodology [15,16].

The Unbiased Evaluation methodology aims to sample numerous subsets of training and test datasets, and then use these datasets to train and test ML models. The model

parameters and variables used in the ML models will be the same as for the most optimal model selected from the Model Construction methodology. The only difference between the models will be the unique subsets of training and test data used for each model. The number of sampled training and test datasets will be numerous in order to produce enough empirical evidence to be able to answer the three RQs despite the data-related challenges discussed in Section 2.3.

Figure 2 illustrates how the two methodologies relate and come together to answer the three RQs.

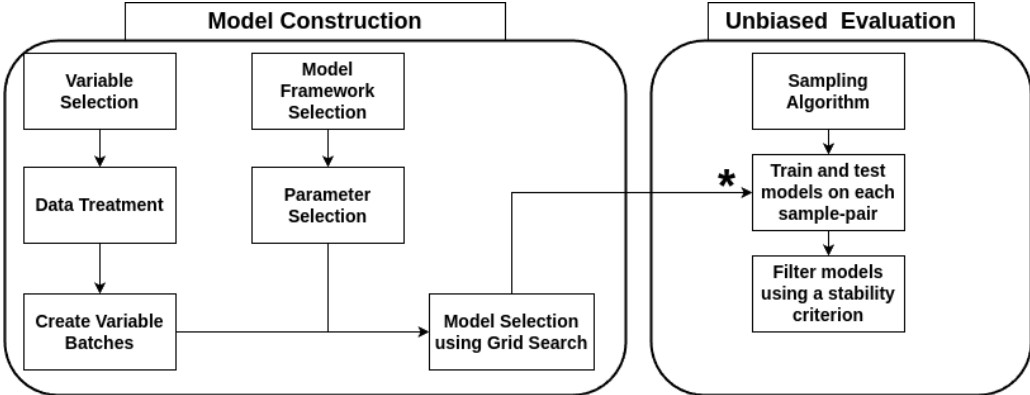

**Figure 2.** Illustration of the relation between the Model Construction methodology developed in previous work and to the Unbiased Evaluation methodology developed in the present work [15,16]. * The parameters of the selected model type in the last step of the Model Construction methodology are used for all machine learning (ML) models that are trained and tested on the training and test data segments, respectively, that are sampled using the sampling algorithm for Unbiased Evaluation.

### 3.1. Processes and Data Sets

The datasets used in the experiments are from four processes located in three different steel plants. Two of the datasets come from two EAFs. The other two datasets come from a VTD and an LRF, respectively. Metadata for the four datasets are shown in Table 1.

**Table 1.** Metadata about the datasets used in the present work. The EAF1 and EAF2 datasets have been used in previous publications [15,16].

| Data Set | Steel Types | Steel Plant Type | Steel Plant | Target Variable | Data (Year) |
|----------|-------------|------------------|-------------|-----------------|-------------|
| EAF1 | Stainless steel | Mini-mill (scrap based) | A | EE consumption | 2017–2019 |
| EAF2 | Ball bearing rings, rods, and engineering steel | Mini-mill (scrap based) | B | EE consumption | 2019–2020 |
| VTD | Wear plates | Integrated | C | End-point steel temperature | 2008–2022 |
| LRF | Wear plates | Integrated | C | End-point steel temperature | 2008–2022 |

The ML models created using datasets EAF1 and EAF2 predicted the EE consumption at the end of each heat in the EAF and have been published previously [15,16]. The ML models created using the datasets from the VTD and the LRF predict the molten steel temperature at the end of the process. The true value of the output variable is the last temperature sample taken in the process for each heating.

### 3.2. Model Construction

Details about the Model Construction methodology used and the corresponding Model Construction for the ML models trained on the EAF1 and EAF2 datasets have been thoroughly explained in previous research [15,16]. Thus, the following sections briefly describe the Model Construction methodology for the models trained on the LRF and VTD

datasets. Both datasets have *not* been involved in previously published research in the field of ML in steel process engineering.

### 3.2.1. Variable Selection

The selected input variables for the ML models for both LRF and VTD datasets were selected based on metallurgical domain knowledge and knowledge about the specific processes in the steel plant from where the data originates.

Table 2 describes the input variables used for the LRF and VTD datasets, respectively. Since it is not possible to know which subset of variables produces the most optimal model with respect to complexity, stability, and predictive performance, several variable batches were created. The variable batches were created based on variable groups, which are presented in Tables A1 and A3 for LRF and VTD in the Appendix A, respectively. The created variable batches will be used as parameters in the grid search for LRF and VTD and are presented in detail in Tables A2 and A4 in the Appendix A.

### 3.2.2. Data Treatment

Since the LRF and VTD processes reside within the same secondary metallurgy station, both processes can be used multiple times for the same heat. However, this is not frequently occurring and these heats were subsequently removed from the dataset used in the present work. In addition, the VTD dataset contains data from all heats that have passed either the VTD only or both the LRF and the VTD. The reason is that it is very rare that heat gets treated by the VTD and not heated in the LRF as the subsequent step. In addition, some raw material types can only be added in the LRF. The LRF dataset, on the other hand, contains data from heats that have only been treated by the LRF. From hereon, the following data treatment was conducted separately for the LRF and VTD datasets and performed using metallurgical process knowledge. This means that the removal of data points was not based on statistical heuristics.

**Table 2.** The variables used to create the ML models for the LRF and VTD datasets. * Boosting is a mode of gas injection that temporarily accelerates the rate of gas injected.

| Data Set (s) | Variable | Unit | Description |
|---|---|---|---|
| LRF/VTD | First measured temperature | °C | The first temperature sample taken in the process |
| LRF/VTD | First predicted temperature | °C | The predicted temperature by the proprietary physico-chemical model at the time of the first temperature sample |
| LRF/VTD | Last measured temperature | °C | The last temperature sample taken in the process. This is the target variable of the developed ML models. |
| LRF/VTD | Last predicted temperature | °C | The predicted temperature by the proprietary physico-chemical model at the time of the last temperature sample |
| LRF/VTD | Process time | min | The time between the first and last measured temperatures |
| LRF/VTD | Start weight | kg | The weight of the molten steel and slag at the time of first temperature measurement |
| LRF/VTD | Addition weight | kg | The weight of all added materials during the process |
| LRF/VTD | Porous plug time | s | The total time the porous plugs were used |
| LRF/VTD | Porous plug volume | Nm$^3$ | The total volume of gas injected through the porous plugs |
| VTD | Number of boosts * | - | The total number of porous plug boosts performed |
| VTD | Vacuum time | s | The total time when the ladle is in vacuum pressure |
| LRF | Lance stirring time | s | The total time the lance were used |
| LRF | Lance stirring volume | Nm$^3$ | The total volume of gas injected through the lance |
| LRF/VTD | Electrode time | s | The total time when the electrodes were powered on |
| LRF/VTD | Electrode energy | kWh | Energy consumed by the heating electrodes |
| LRF/VTD | Ladle empty time | min | For how long the ladle was empty before the tapping from the BOF to the ladle |
| LRF/VTD | Ladle trips | - | Number of heats the ladle has been used since the last ladle re-bricking |

Any negative values in the datasets were removed. This is straightforward since the values for all variables in the dataset as shown in Table 2 can only be either zero or positive.

Furthermore, the time between the first and last temperature samples must be reasonable. If the time between the samples is too short, then the temperature prediction becomes trivial. On the other hand, if the time between the samples is too long, then there is an increased risk of including heats that have been exposed to rare production delays and irregularities. The predicted and measured temperature values were filtered between the lower bound and upper bound of possible temperature values. Values representing the empty time of the ladle that was similar to the time of major maintenance overhauls were also removed. This was carried out to ensure that heats produced directly after the major maintenance overhauls were not included in the dataset. These heats are often challenging to process and therefore non-representative of regular production. Lastly, any rows where the lance stirring time, porous plugs time, electrode time, or vacuum time were reported to be longer than the process time were removed.

The aforementioned data treatment steps had the following impacts on the LRF and VTD datasets. For the LRF dataset, the number of data points was reduced from 32,890 to 29,033, representing a reduction of 11.7%. The number of data points in the VTD dataset, on the other hand, was reduced from 45,714 to 43,294, representing a reduction of 5.3%.

### 3.2.3. Model Framework and Parameter Selection

For the sake of comparability, the model framework used to create the ML models for the LRF and VTD datasets will be the same as the model framework used for the models created for the EAF1 and EAF2 datasets. Hence, the ANN model framework will be used. The selected values for the parameters specific to the ANN model framework are shown in Table 3.

**Table 3.** The parameters specific to the Artificial Neural Network (ANN) model framework that is used during the grid search for both the LRF and VTD datasets. The total number of model-specific parameter combinations is 96 ($2 \cdot 3 \cdot 16$).

| Parameter | Description | Values | # |
|---|---|---|---|
| Activation function | The activation function influences the training phase since the updating step of the gradient descent algorithm will be different. | $[tanh, logistic]$ | 2 |
| Learning rate | The learning rate adjusts the step size taken in loss space during gradient descent and therefore influences both the predictive performance of the end model and the training time. | $[0.001, 0.01, 0.1]$ | 3 |
| Topology of the hidden nodes | The number of hidden layers and hidden nodes in each layer correlates with a higher model complexity. On the other hand, a more complex model has the ability to learn more complicated relations between variables in the data set. | $(z)$ or $(z, z)$ where $z \in 1, 4, \ldots, 22$ | 16 |

The rest of the available parameters were left as default per the documentation of the software package in the Python programming language which was used to create the models [31].

### 3.2.4. Grid Search

A total of 96 (Table 3) parameters, together with a total of 48 (Tables A2 and A4) variable batches (VB), represent the total number of model types that will be created in the grid search for the LRF and VTD datasets. For LRF, the total number of model types is 1536, while the number of model types for VTD is 3072. Grid search is a well-known and frequently used framework that enables the modeler to find the parameters that create the most optimal model. In the Model Construction methodology, the goal is to find and select the most optimal model type with respect to stability, complexity, and predictive performance.

In the present work, model type refers to a model with a specific set of parameters and data used to train and test the model. A model instance, on the other hand, is a

specific instance of a specific model type. By analyzing the predictive performance of each model instance of a specific model type, it is possible to evaluate the stability of the model type. In the present work, 10 instances of each model type will be created. In addition, the random state of each of the 10 model instances will be held fixed for all model types. There are two important reasons for this. First, keeping a random state constant ensures that the 10 model instances can be reproduced given the same dataset and model parameters. Second, the predictive performance between the model types should not depend on the randomness introduced by the ANN algorithm. It should only be dependent on the sampled training dataset and test dataset. By fixing the random state, the randomization procedure will be identical for all models trained using that specific random state. The values of the random states are immaterial. The important aspect is that each of the 10 values is unique and that they are kept fixed for all model types in the experiments. The random state is defined in the documentation for the software package used to create the models [31].

The selected random state values, $Z_s$, in order of model instance are:

$$Z_s \in [41, 901, 203, 61, 10, 1001, 53, 201, 702, 311] \tag{1}$$

### 3.2.5. Predictive Performance Metric

The predictive performance metric used in the present work is the adjusted-$R^2$, which is also known as the adjusted coefficient of determination. It is defined as:

$$R^2_{adj} = 1 - (1 - R^2)\frac{n-1}{n-p-1} \tag{2}$$

where $R^2$ is the regular $R^2$ metric, $p$ is the number of input variables used by the ML model, and $n$ is the number of data points in the dataset under consideration. The reason that the adjusted R-squared metric is used is because it enables the comparison between ML models that use a varying number of data points and input variables [32]. In addition, the adjusted R-squared metric was used as a predictive performance metric in previous work that reported the most optimal models for EAF1 and EAF2 [15,16]. The present work does not aim to modify the Model Construction methodology.

Since 10 model instances will be created for each model type in order to evaluate its stability, the following statistical quantities, derived from the $R^2_{adj}$-values of the 10 model instances, will also be used:

- $\bar{R}^2_{tr}$—the mean adjusted-$R^2$ on training data for a model type.
- $\bar{R}^2_{te}$—the mean adjusted-$R^2$ on test data for a model type.
- $R^2_{min,te}$—the minimum adjusted-$R^2$ on test data for a model type.
- $R^2_{max,te}$—the maximum adjusted-$R^2$ on test data for a model type.

### 3.2.6. Model Selection

As previously mentioned, the most optimal model is a model that is stable, achieves high predictive performance, and has low complexity. Since these three aspects are difficult to achieve simultaneously, the following model selection criteria were used [15].

1. Remove any model that does not satisfy the following condition: $R^2_{max,te} - R^2_{min,te} \leq 0.05$. This ensures that models passing the criterion are stable.
2. Sort the model types on decreasing $\bar{R}^2_{te}$ of the 10 model instances. This ensures that the model type with the highest predictive performance is selected among the stable model types.
3. Select the first model type in the resulting list.

The metadata showing the variable batch, model parameters, and predictive performance for each of the most optimal model types for the four datasets are shown in

Table 4. The selected model type for each of the four datasets will be used in the Unbiased Evaluation methodology, which will be explained in detail in the next section.

**Table 4.** Relevant metadata about the most optimal model types for the four datasets. The metadata from the ML models trained and evaluated on data from EAF1 and EAF2 are provided from previous work [15,16]. Details about the parameters specific to the ANN algorithm are explained in the documentation of the software used [31].

| Data Set | Variable Batch | Activation Function | Learning Rate | Topology of the HIDDEN Layers | $\bar{R}^2_{te}$ | Training Data Points | Test Data Points |
|---|---|---|---|---|---|---|---|
| EAF1 | [15] | *logistic* | 0.001 | (11) | 0.731 | 10,966 | 362 |
| EAF2 | [16] | *tanh* | 0.01 | (29) | 0.490 | 2571 | 187 |
| VTD | VB11 | *tanh* | 0.001 | (22, 22) | 0.679 | 28,255 | 286 |
| LRF | VB7 | *logistic* | 0.001 | (16) | 0.634 | 42,861 | 433 |

### *3.3. Unbiased Evaluation*

### 3.3.1. Data Sets and Model Parameters

There are two aspects that relate to the trade-off between the number of numerical experiments and the empirical strength of the experimental results that have to be discussed. First, the model complexity must be high enough to encompass the complexity of the data that is used to adapt the model. Second, the selected variables may not be the subset of variables that produce the model with the highest predictive performance on the test data for all subsets of training and test data produced by the sampling algorithm. The following paragraphs motivate why the model parameters and input variables of the most optimal models from the Model Construction methodology will be used in the models created in the Unbiased Evaluation methodology. **Model Complexity:** It is important that the model complexity is high enough to enable the model to accurately adapt its parameters to the data used to train the model. The results presented in Table 4 indicate the optimal complexity for each model needed to create a stable model with the highest predictive performance on test data for each of the datasets. It is reasonable to assume that the data complexity is larger for a dataset with a larger number of data points compared to one that has a relatively few number of data points. This is because a larger dataset represents the process conditions over a longer period of time. The process conditions always change over time due to varying demand for certain steel grades, general process development, and wear of process equipment. This is why the model complexity, i.e., number of hidden nodes, number of hidden layers, and number of input variables will be selected based on the best performing model from the Model Construction methodology, which used the complete dataset. In addition, in the Model Construction methodology, the fraction of training data accounted for between 93% and 99% of the number of data points in the complete datasets. This means that the parameters of the models reported in Table 4 were adapted on training sets that encompass the overhanging majority of the complete dataset. Hence, the assumption can be made that the model parameters specified in Table 4 produce models that are complex enough to be used for any subset of their corresponding complete data sets. **Selected Variables:** It is not possible to know, a priori, if the selected variables will be the best performing subset of variables for all subsets of training and test data collected from the sampling algorithm. Ideally, one should conduct a grid search to select the best input variable subset and the best model-specific parameters for each subset of training data and test data. However, this introduces a significant number of additional parameter combinations for an already large number of model instances that are required for the Unbiased Evaluation methodology experiments. The current number of model instances per dataset is in the order of $10^4$ to $10^5$. A meaningful grid search on each subset would require at least $10^7$ model instances on each dataset. One approach could be to use all available variables gathered from the process systems by ignoring the principle of

parsimony. However, this is ill-advised since the resulting model will contain variables that are highly correlated. This, in turn, diminishes the trustworthiness of the model because these variables will then share feature importance, which results in inaccurate feature importance values [33]. Another approach could be to use the same variables as the best model determined on the complete dataset. While it is not possible to know, with certainty, that these variables will be the best performing for each subset of training data and test data, they have been selected using domain expertise. The advantage of this approach is two-fold. First, the models must make sense from a domain-specific point of view to be trusted among domain experts. Second, the number of selected variables will be fewer than the available variables in the processing system since the domain experts know, from experience, which process variables significantly influence the target variable. Based on the above reasoning, the specific input variables used for the most optimal model for each of the four datasets, as reported in Table 4, will be used as input variables for the models that will be created in the Unbiased Evaluation methodology.

This means that the exact same model parameters and input variables that were used to create the most optimal models in the Model Construction methodology will be used in the Unbiased Evaluation methodology. The only differences between the model types are the start points, end points, and total number of data points in the training and test datasets, respectively.

### 3.3.2. Sampling Algorithm for Unbiased Evaluation

It is important to conduct the selection of the start point of the training data and the number of training data and test data points in an unbiased manner. In essence, this allows the resulting datasets to become the basis of which to answer the three RQs of the present work. Furthermore, the nature of steel processes demands that the ML model is always evaluated on test data that follow in chronological order after the training data. Hence, Algorithm 1 will be used to achieve unbiased sampling start point of the training data, the number of training data points, and the number of test data points. Naturally, the start of the test data is the data point that occurs right after the last data point in the training data.

---

**Algorithm 1:** Sampling algorithm

- Define $N$ as the total number of data points in the dataset.
- Define $min_{tr}$ as the minimum number of training data points allowed in any sample
- Define $min_{te}$ as the minimum number of test data points allowed in any sample
- Select the number of samples, $S_n$.
- For each $s \in S_n$, do:
    1. Sample the start point of the training data, $p_{start}^{tr}$ from the uniform distribution $[\, 0, N - (min_{tr} + min_{te})\,]$
    2. Sample the size of the training data, $s_{tr}$, set from the uniform distribution $[\, min_{tr}, N - (p_{start}^{tr} + min_{te})\,]$
    3. Sample the size of the test data, $s_{te}$, from the uniform distribution $[\, min_{te}, N - (p_{start}^{tr} + s_{tr})\,]$
    4. Calculate the end point of the training data as $p_{end}^{tr} = p_{start}^{tr} + s_{tr}$
    5. Calculate the start point of the test data as $p_{start}^{te} = p_{start}^{tr} + s_{tr} + 1$
    6. Calculate the end point of the test data as $p_{end}^{te} = p_{start}^{tr} + s_{tr} + 1 + s_{te}$
    7. Add each randomized value and calculated value into corresponding lists
- Append the lists into a table which will be used to retrieve the specific training and test data points for the next step of the experiments.

---

One sample from the sampling algorithm is illustrated in Figure 3.

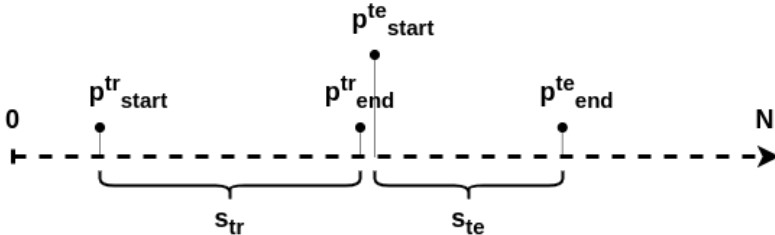

**Figure 3.** Illustration of one generated sample of start point of the training data, number of training data points, and number of test data points, using the sampling algorithm.

To run the sampling algorithm, three parameters have to be pre-specified. These are the number of samples, $S_n$, the minimum number of training data points, $min_{tr}$, and the minimum number of test data points, $min_{te}$. In the present work, the selection of $S_n$ was set equal to the number of available data points in the dataset, $N$. This gives a high probability that each possible start point of the training data gets sampled once. This, in turn, ensures a high diversity with respect to the selected training data points and test data points since these are sampled from distributions that are dependent on the start point of the training data, $p_{start}^{tr}$. $min_{tr}$ was always set to 100 so that the model attains some relevant experience prior to predicting test data. To ensure that the predictive performance on test data for each model is valid, $min_{te}$ was also set to 100.

The distributions of the sampled and calculated values are shown in Figure 4. The types of distributions are a direct outcome of the requirements that are set on the relationship between the training and test data as well as the total available number of samples. For example, it is obvious that samples that have approximately 200 training data points are more common than samples that have approximately 10,000 training data points. This is because the number of different ways to sample segments of 200 training data points is significantly larger than the corresponding 10,000 training data points. The reason is that the sampled start point of the training data is sampled randomly and, per definition, demarcates the upper bound of the number of training data points to sample. Analogously, the number of test data points is dependent on both the start points of the training data and the number of training data points.

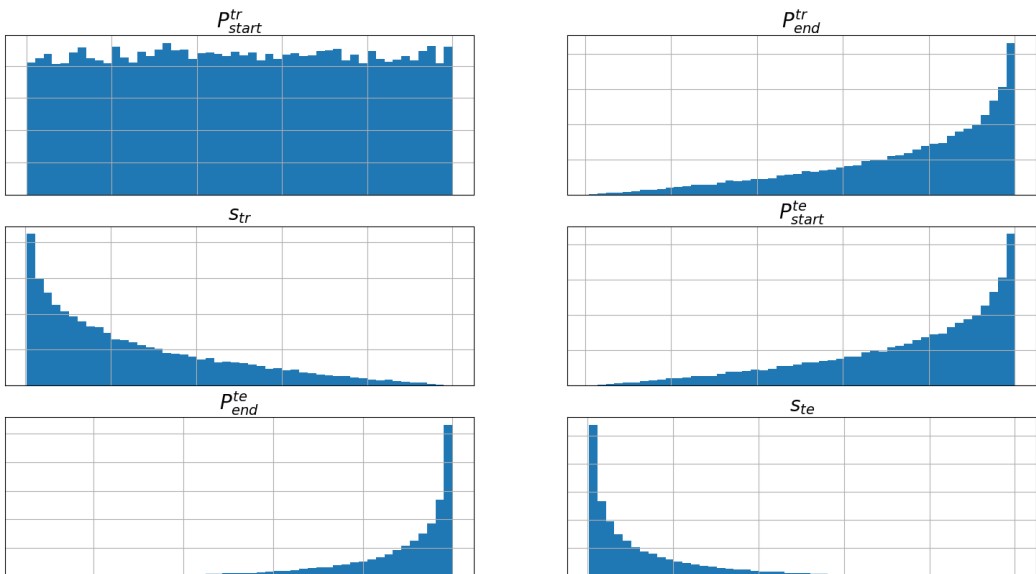

**Figure 4.** Illustration over the distribution of the parameters $p_{start}^{tr}$, $s_{tr}$, and $s_{te}$, and the calculated values $p_{end}^{tr}$, $p_{start}^{te}$ and $p_{end}^{te}$, using the sampling algorithm.

### 3.3.3. Filtering Stable Models

Since the ANN algorithm introduces randomness, there is a probability that instances of the same model type will have different predictive performance values. To account for this, the same approach reported in the Model Construction methodology will be used. In short, 10 model instances will be created for each model type and the difference between the maximum and minimum adjusted-$R^2$ metric for the 10 models will be calculated. While all models have to be retained for visualization, all model types that fail to satisfy the criterion, $R^2_{max,te} - R^2_{min,te} \leq 0.05$, will be flagged so that they can be omitted from the analysis if needed. This criterion is the same as the one used to remove unstable models in the Model Construction methodology. Furthermore, the same random state values as the ones used in the Model Construction methodology will be used. Hence, the total number of model instances will be 10 times the number of model types, which is determined using the sampling algorithm presented in Section 3.3.2. This filtration criterion ascertains that the models satisfy the stability requirement in the definition of upper-bound predictive performance as explained in Section 2.1: *The model producing the upper-bound predictive performance must be stable, i.e., the predictive performance must not be the result of a randomization procedure within the ML model algorithm.* Hence, the filtered models become the basis of the analysis answering the three RQs in the present work.

### 3.3.4. Statistical Analysis of the Stable Models

To analyze the predictive performance of the stable models, several statistical tools and metrics will be used. As basis of the analysis, $\bar{R}^2_{te}$ will be plotted against $s_{tr}$, $s_{te}$, and $P^{te}_{start}$, all of which were defined in Sections 3.2.5 and 3.3.2.

Furthermore, two statistical analysis methods will be used to enable an easier interpretation of the scatter plots. The first method is known as Pearson correlation which quantifies the linear relationship between two random variables through a test of significance [34]. The Pearson correlation can assume values between $-1$ and 1, where a positive value indicates that the variables are positively correlated and a negative value indicates that the variables are negatively correlated. A correlation value of 0 means that the variables lack correlation. The test of significance produces a value known as the *p*-value, which measures the probability of obtaining the specific correlation value, assuming that the null hypothesis is true. The null hypothesis for the Pearson correlation test is that the correlation coefficient *is not* significantly different from zero. A low *p*-value, usually $p \leq 0.05$, means that the observed correlation value is statistically significant. The second method is the moving average of the values on the *y*-axis per a determined interval on the *x*-axis. The *y*-axis will represent $\bar{R}^2_{te}$ and the *x*-axis will represent $s_{tr}$, $s_{te}$, or $P^{te}_{start}$. The moving average curve in the present work will use an interval of size 200. This value was not selected arbitrarily, rather it was selected based on how well it illuminates the local trends in the graphs. The calculation for a specific interval, *j*, of the 200-moving average curve is as follows:

$$M^{avg,j}_{200} = \frac{\sum_{i=1}^{n_j} y_i}{n_j} \tag{3}$$

where $i \in 1, 2, \ldots, n_j$ and $n_j$ is the total number of stable models in a specific interval, $j \in 1, 2, \ldots, N_v$. $N_v$ is the total number of 200-value intervals.

It is important to reiterate that the Pearson correlation statistic considers all data points to produce a single value while the 200-value moving average calculates its average based on the values present in trailing 200-value segments. Hence, these two metrics show a global and local relationship between two variables, respectively.

## 4. Results and Discussion

The sampling algorithm results for the four datasets are shown in Table 5. Per the definition of the sampling algorithm for Unbiased Evaluation and the selected total number of models, the total number of models equals the number of data points. However, for EAF1,

EAF2, and LRF, the algorithm produced several training and test data intervals with the same length and start positions. Since these samples are identical, they were removed.

**Table 5.** Sampling algorithm results for the four datasets. The stability filter used was $R^2_{max,te} - R^2_{min,te} \leq 0.05$ which is the same stability filter used in the process of selecting the most optimal model with respect to stability, predictive performance, and complexity.

| Data Set | No. Data Points | No. Models | No. Models after Stability Filter | Removed by Stability Filter |
|---|---|---|---|---|
| EAF1 | 11,328 | 11,324 | 1048 | 90.7% |
| EAF2 | 2758 | 2757 | 95 | 96.6% |
| VTD | 29,033 | 29,033 | 24,560 | 15.4% |
| LRF | 43,294 | 43,292 | 33,888 | 21.4% |

It is evident that the fraction of models that are removed by the stability filter is significantly larger for the EAF datasets. This agrees well with the known behavior of the EAF that is consequently reflected in the data. Frequent changes in charged scrap in the EAF, which also affect the EE consumption, make it challenging to consistently produce stable models since the frequent changes will be interpreted as noise. The number of models passing the stability filter for EAF2 is low compared to the other processes. While 95 models can be used to draw conclusions, the empirical evidence of these conclusions will not be as strong as for the other datasets, which have between 1048 and 33,888 models passing the stability filter. The following graphs (Figures 5–7) were created using the results from the experiments that were filtered using $R^2_{max,te} - R^2_{min,te} \leq 0.05$. Each graph contains a line representing a 200-value moving average over the *x*-axis. Furthermore, each graph also has a corresponding Pearson correlation test calculated using the variables represented by the *x*-axis and *y*-axis.

*4.1. Research Question 1*

It is possible to determine the number of training data points that are needed to create an ML model with near-upper-bound predictive performance on test data by observing the highest $\bar{R}^2_{te}$-value. However, the highest $\bar{R}^2_{te}$-value may reside in a region of $s_{tr}$ where $\bar{R}^2_{te}$ has large variance indicating that $\bar{R}^2_{te}$ could be lower than the near-upper-bound predictive performance for a similar number of training data points. Thus, a relatively low variance of $\bar{R}^2_{te}$ at the highest $\bar{R}^2_{te}$-values will also be an indicator when answering this RQ. In addition, the maximum of the 200-value moving average curve shows what the $\bar{R}^2_{te}$-value is expected to be by taking into account all $\bar{R}^2_{te}$-values for every 200-value segment of $s_{tr}$.

For the EAF1 dataset, as observed in Figure 5, the highest $\bar{R}^2_{te}$ occurs at around 10,500 training samples where it varies between 0.67 and 0.78. $\bar{R}^2_{te}$ is almost as high around 2000 training samples but the variance is considerably larger since $\bar{R}^2_{te}$ varies between 0.23 and 0.76. The Pearson correlation value of 0.51 and the 200-value moving average curve provide further evidence that the highest $\bar{R}^2_{te}$ is expected to occur when $s_{tr}$ is around its maximum value. Since the Pearson correlation value is positive, $\bar{R}^2_{te}$ increases linearly with increasing $s_{tr}$. The 200-value moving average curve is also steadily increasing, albeit irregularly, and peaks at around 10,500 training samples.

For the EAF2 dataset, the highest $\bar{R}^2_{te}$ occurs when $s_{tr}$ is around 1700 and 2200 where the variance is slightly higher in the former segment. The Pearson correlation value is $-0.29$, which indicates that $\bar{R}^2_{te}$ will decrease with higher $s_{tr}$. The 200-value moving average line adds additional empirical evidence that $\bar{R}^2_{te}$ decreases with higher $s_{tr}$.

Two distinct regions can be observed for $\bar{R}^2_{te}$ produced by the models on the LRF dataset. The first region exists when $s_{tr}$ is between 100 and 13,800 where $\bar{R}^2_{te}$ between 0.40 and 0.93. Simultaneously, the second region extends from 13,800 to the maximum number of possible $s_{tr}$ where $\bar{R}^2_{te}$ varies between 0.37 and 0.81. One possible explanation for the high variance of $\bar{R}^2_{te}$ for the LRF dataset is that the LRF production pattern changed due to

changes in steel types produced, which in turn is due to changes in demand during the past 14 years. The Pearson correlation value is 0.03, which indicates that a higher $s_{tr}$ barely increases the $\bar{R}^2_{te}$ of the created models. However, the 200-value moving average line peaks at an $s_{tr}$-value of 4500 and decreases significantly after a $s_{tr}$-value of 27,000. Again, this indicates a difference between the global and local relationship between $\bar{R}^2_{te}$ and $s_{tr}$.

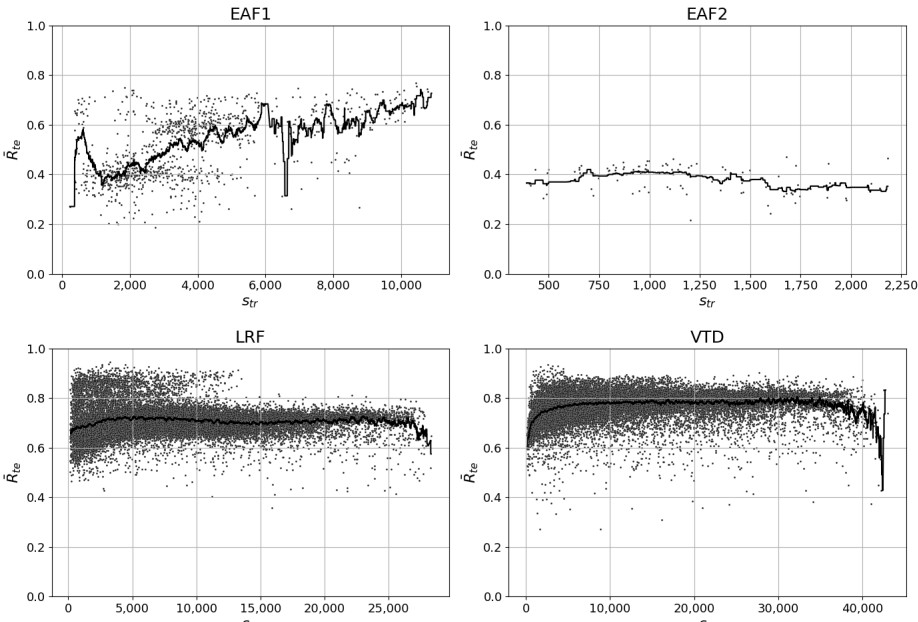

**Figure 5.** The mean R-squared on the test data ($\bar{R}^2_{te}$) plotted against the number of data points in the training set ($s_{tr}$) for all randomized samples whose model types satisfy $R^2_{max,te} - R^2_{min,te} \leq 0.05$. Pearson correlation tests (statistic, *p*-value). - **EAF1:** (0.51, 0.0) - **EAF2:** (−0.29, 0.004) - **LRF:** (0.03, 0.0) - **VTD:** (0.22, 0.0).

A similar reasoning can be conducted for the VTD results as for the LRF results. However, the VTD has only one main region but the variance of $\bar{R}^2_{te}$ is even larger, i.e., between 0.28 and 0.93. Furthermore, the Pearson correlation value is 0.22 while the peak of the 200-value moving average line occurs when $s_{tr}$ is around 30,000. However, the value of the 200-value moving average line does not increase significantly after the $s_{tr}$-value exceeds 10,000. The most prominent change in the average line occurs when the $s_{tr}$-value is between 100 and 10,000. Lastly, the moving average decrease at the end is more prominent than the corresponding curve for LRF. The reason why the curve jumps back from the decrease at the end of the curve is because the last point is the only point available in the 200-value segment used to calculate that specific 200-value moving average value.

The decrease in the moving average curve for the LRF and VTD means that almost all available training data used by the models makes the models perform worse on the test data compared to when the models are trained on other segments of training data. If a model is trained on almost all available training data then it will also be tested on a subset, or the full set, of remaining data points according to the sampling algorithm described in Section 3.3.2. Since the predictive performance decreases for this segment of test data points, the distribution change reflected in this segment makes the test dataset different from the training dataset to such an extent that it warrants a significant decrease in $\bar{R}^2_{te}$. Significant changes in the process layout were performed in steel plant C during the year 2021. The datasets from both LRF and VTD represent production from 2008 until the middle of 2022. Hence, the aforementioned change is a reasonable explanation for this decrease in $\bar{R}^2_{te}$.

The challenge in answering this RQ is illuminated by the fact that the three indicators used, i.e., the highest $\bar{R}^2_{te}$-values, a relatively low variance of $\bar{R}^2_{te}$ at the highest $\bar{R}^2_{te}$-values,

and the maximum of the 200-value moving average curve, do not always coincide in the graphs. Consequently, it is here where where the utility of the proposed method shows itself since it facilitates the analysis of the models to enable stakeholders to deduce the number of training data points required to create a model with near-upper-bound predictive predictive performance on test data.

### 4.2. Research Question 2

Determining the near-upper-bound predictive performance on test data could be performed by analyzing the same figure that was used to answer RQ1, i.e., Figure 5. However, this figure does not provide information on which segment of test data produces the near-upper-bound predictive performance. On the other hand, Figure 6 shows the relationship between $\bar{R}^2_{te}$ and $P^{te}_{start}$.

For EAF1, the model with the highest $\bar{R}^2_{te}$-value, i.e., 0.77, is located where $P^{te}_{start}$ is above 11,000, which is almost at the end of the complete dataset. In addition, the Pearson correlation value is 0.73 and the trend of the 200-value moving average curve is positive when $P^{te}_{start}$ is above 11,000. However, some caution should be exercised on these results since it is not possible to know if this trend in predictive performance will continue in subsequent produced heats. After all, the $\bar{R}^2_{te}$-values when $P^{te}_{start}$ is below 9700 varies between 0.2 and 0.55. It is only after the abrupt jump, which occurs when $P^{te}_{start}$ is around 9800, that the $\bar{R}^2_{te}$-values concentrate between 0.55 and 0.77. In addition, the reason behind this sudden improvement in $\bar{R}^2_{te}$ is not known.

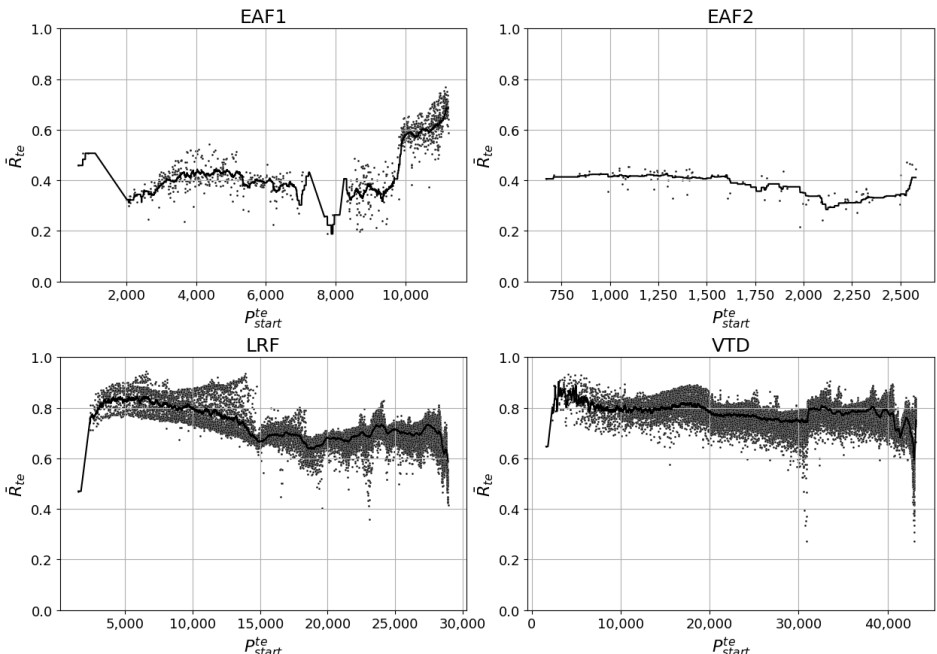

**Figure 6.** The mean R-squared on the test data ($\bar{R}^2_{te}$) plotted against the start of test set ($P^{te}_{start}$) for all randomized samples whose model types satisfies $R^2_{max,te} - R^2_{min,te} \leq 0.05$. Pearson correlation tests (statistic, *p*-value). **EAF1:** (0.73, 0.0) - **EAF2:** (−0.52, 0.60) - **LRF:** (−0.46, 0.0) - **VTD:** (−0.27, 0.0).

Like EAF1, $\bar{R}^2_{te}$-values for EAF2 are highest when $P^{te}_{start}$ is at the end of the dataset, i.e., when $P^{te}_{start}$ is above 2500. The 200-value moving average curve indicates that $\bar{R}^2_{te}$ recovers from its lows, occurring at a $P^{te}_{start}$-value of 2120, and ends at its highest when $P^{te}_{start}$ is between 2500 and 2678. Hence, the near-upper-bound predictive performance on test data for EAF2 is a $\bar{R}^2_{te}$-value 0.44.

The results of the LRF dataset show a steady decrease in $\bar{R}^2_{te}$ for test data from earlier segments to the last segment of the dataset. This is shown by both the 200-value moving average curve and the Pearson correlation value of −0.46. While the near-upper-bound predictive performance is a $\bar{R}^2_{te}$-value of 0.93, it is not expected to reoccur soon considering

that the long-term effects of the previously mentioned production overhaul performed in steel plant C has not yet been established. Furthermore, both the negative Pearson correlation value and the negative trend of the 200-value moving average curve illuminate a continuing decrease in predictive performance. Since the LRF and VTD processes are located in the same steel plant, similar reasoning can be used for the results from the VTD process. Here, the 200-value moving average also trends downward in the last segment of the dataset and the Pearson correlation value is $-0.27$. The near-upper-bound predictive performance for the VTD is a $\bar{R}^2_{te}$-value of 0.92.

It should be noted that there are very few stable models when $P^{te}_{start}$ occurs at the earliest segment of the dataset for all four processes. The corresponding graph, i.e., Figure A1 in the Appendix A, contains the $\bar{R}^2_{te}$-values for all models and illuminates the minimum number of training data points needed to create stable models. If $P^{te}_{start}$ occurs early in the dataset, then $s_{tr}$ will also be low per the definition of the sampling algorithm.

*4.3. Research Question 3*

By observing Figure 7, it is possible to understand how long a model can be used before its predictive performance starts to decrease. Here, the start point of a decrease in the 200-value moving average curve will be determined as the $s_{te}$-value that indicates how long the model can be used.

For EAF1, $\bar{R}^2_{te}$ decreases significantly immediately following the start of the 200-value moving average curve, i.e., when $s_{te}$ is 200. This means that models predicting the EE of the EAF1 should be retrained after predicting on 200 test data points.

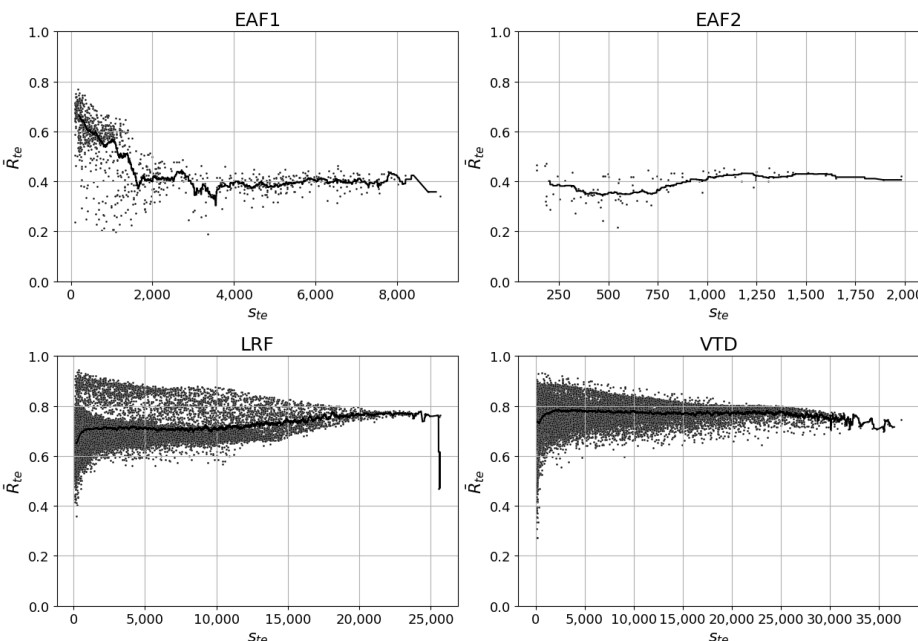

**Figure 7.** The mean R-squared on the test data ($\bar{R}^2_{te}$) plotted against the number of data points in the test set ($s_{te}$) for all randomized samples whose model types satisfy $R^2_{max,te} - R^2_{min,te} \leq 0.05$ Pearson correlation tests (statistic, *p*-value). - **EAF1:** $(-0.64, 0.0)$ - **EAF2:** $(0.45, 0.0)$ - **LRF:** $(0.19, 0.0)$ - **VTD:** $(0.05, 0.0)$.

For EAF2, the models with the highest $\bar{R}^2_{te}$-values are the ones that have $s_{te}$-values below 200. However, the 200-value moving average curve indicates that a higher $s_{te}$-values leads to a slightly higher $\bar{R}^2_{te}$-value. This increase in $\bar{R}^2_{te}$, albeit low, is contradictory to established ML knowledge since a decrease in $\bar{R}^2_{te}$ is expected if the model has been applied on more test data points, i.e., when the $s_{te}$-value is higher. Data points that are chronologically adjacent are more similar than data points that are chronologically far

apart, which is partly explained by changes in the steel process under study. The Pearson correlation value of 0.45 provides further evidence that $\bar{R}^2_{te}$ increases when $s_{te}$ increases.

For LRF, the models with the highest $\bar{R}^2_{te}$-values are the ones that have $s_{te}$-values in the lower region. However, this region also produces models that have the largest range of possible $\bar{R}^2_{te}$-values. The range of $\bar{R}^2_{te}$-values are between 0.35 and 0.93. The range decreases significantly when $s_{te}$ approaches 15,000 and even more so beyond 20,000. However, the similarity between test datasets that approaches the maximum number of possible test data points is large, per definition of the sampling algorithm. Therefore, it is expected that the variance of $\bar{R}^2_{te}$ decreases when $s_{te}$ approaches its maximum value. Contrary to expectations and similarly to EAF2, both the 200-value moving average curve and the Pearson correlation value of 0.19 illustrate that the $\bar{R}^2_{te}$ increases when $s_{te}$ increases. Hence, the LRF model can be used for 24,000 heats before its predictive performance starts to deteriorate. The severe sudden decrease in the moving average curve at its end is because only one stable model with a relatively low $\bar{R}^2_{te}$-value exists in the last 200-value interval.

The results from the VTD models are similar to those for the LRF models, reflecting the fact that these two processes have similar process characteristics and reside in the same steel plant. The Pearson correlation value is 0.05 and the 200-value moving average curve increases. However, the moving average curve only increases when the $s_{te}$-value is between 100 and 2000. Beyond an $s_{te}$-value of 2000, the moving average curve slowly decreases from 0.79 to 0.72.

Table 6 summarizes the answers to the three RQs as discussed in this section.

**Table 6.** Summary and answer to the three research questions (RQs) for the four data sets.

**RQ1: How many training data points are needed to create a model with near upper-bound predictive performance on test data?**

| Process | $s_{tr}$ | % of complete dataset | Pearson | *p*-value |
|---|---|---|---|---|
| EAF1 | 10,500 | 92.7% | 0.51 | 0.0 |
| EAF2 | 2200 | 79.8% | −0.29 | 0.004 |
| LRF | 4500 | 15.5% | 0.03 | 0.0 |
| VTD | 30,000 | 69.3% | 0.22 | 0.0 |

**RQ2: What is the near-upper-bound predictive performance on test data?**

| Process | $\bar{R}^2_{te}$ | Pearson | *p*-value |
|---|---|---|---|
| EAF1 | 0.77 | 0.73 | 0.0 |
| EAF2 | 0.44 | −0.52 | 0.60 |
| LRF | 0.93 | −0.46 | 0.0 |
| VTD | 0.92 | −0.27 | 0.0 |

**RQ3: For how long can a model be used before its predictive performance starts to decrease?**

| Process | $s_{te}$ | Pearson | *p*-value |
|---|---|---|---|
| EAF1 | 200 | −0.64 | 0.0 |
| EAF2 | 200 | 0.45 | 0.0 |
| LRF | 24,000 | 0.19 | 0.0 |
| VTD | 2000 | 0.05 | 0.0 |

## 5. Conclusions

In the process of answering the three RQs, several other observations with corresponding conclusions could be made. First, the removal of 91% and 97% of the EAF1 and EAF2 models, respectively, using the stability filter provides evidence that EAF datasets are noisier than LRF and VTD datasets. This agrees well with practical experience and is likely due to frequent changes in the amount and variety of charged scrap types as well as other factors influencing the process. The corresponding percentages for the LRF and VTD datasets were 15% and 22%, respectively. Second, the commonly used 80/20 split of all data

into a training set and test set, respectively, is not a one-size-fits-all approach for training and evaluating ML models applied in the context of steel processes. The training data needed for the four processes varied between 15.5% and 92.7% of the complete datasets. To find the number of training data points needed, an analysis using the proposed sampling algorithm could be used. One should not use all data points unless necessary. For example, if the number of relevant training data points is less than the total number of data points, then the most recent segment of data should be used as training data. Third, contrary to the established ML knowledge, there was higher predictive performance of test data with higher test data samples for the EAF2 and LRF models. It is commonly expected that the predictive performance of test data decreases the longer a model is used. Lastly, the predictive performance on test data can be expected to have high variability both within specific segments of test data and between segments of test data. To understand the reason behind this variability, one could use historical reports of changes conducted within the steel plant and the specific steel process in conjunction with the dataset presented in chronological order. These historical reports could, for example, explain when maintenance was conducted or an SOP was changed. Ideally, however, the steel industry ought to invest in a more systematic way of tracking the changes in the data imposed by changes in the process itself or in the steel plant where the process resides.

The concluding answers to the three RQs are as follows:

- *RQ1: How many training data points are needed to create a model with near-upper-bound predictive performance on test data?* The fraction of training data points of the total number of data points needed to create a model with near-upper-bound predictive performance on test data varies between 15.5% and 92.7% for the four datasets. Hence, using as much training data as possible is not always required to create a model with near-upper-bound predictive performance on test data. The number of training data points required should be based on an analysis using the proposed sampling algorithm.

- *RQ2: What is the near-upper-bound predictive performance on test data?* This RQ can easily be answered by observing the highest $\bar{R}^2_{te}$ in any of the provided graphs. For the processes in order of EAF1, EAF2, LRF, and VTD, the near-upper-bound $\bar{R}^2_{te}$ were 0.77, 0.44, 0.93, and 0.92. However, it is of value to determine if the near-upper-bound predictive performance is sustained for heats following in chronological order of the present dataset. Since the 200-value moving average curve for EAF1 and EAF2, at their ends, have an upward trajectory in Figure 6, one could assume the $\bar{R}^2_{te}$ will be at least as high for future heats. However, for EAF1 there is a sudden increase in $\bar{R}^2_{te}$ at the end of the dataset which warrants some caution about whether this predictive performance can be sustained for future heats. For the LRF and VTD datasets, the 200-value moving average curves decrease significantly when $P^{te}_{start}$ approaches the end of the respective dataset. The reason is because of the production overhaul that occurred during the year 2021 in steel plant C, which made the latest segment of the dataset different from the earlier segments. Since the long-term effects of this change have not yet been established, it is difficult to expect that the current near-upper-bound predictive performance value will reoccur in the near future.

- *RQ3: For how long can a model be used before its predictive performance starts to decrease?* For the EAF processes, the predictive performance of the models starts to decrease after 200 predictions on test data. For the VTD process, the model can be used for 2000 predictions. Surprisingly, the LRF model can be used to predict the end-point temperature of 24,000 heats. Assuming a yearly production between $10^3$ and $10^4$ illuminates the varying frequency of model retraining for the four processes. In addition, $\bar{R}^2_{te}$ moving average curve increases from 0 to 24,000 pointing to the fact that $\bar{R}^2_{te}$ improves with larger test datasets. This is contrary to established ML modeling knowledge.

**Author Contributions:** Conceptualization, L.S.C. and P.B.S.; methodology, L.S.C. and P.B.S.; software, L.S.C.; validation, L.S.C. and P.B.S.; formal analysis, L.S.C.; investigation, L.S.C.; resources, L.S.C.; data curation, L.S.C.; writing—original draft preparation, L.S.C.; writing—review and editing, L.S.C. and P.B.S.; visualization, L.S.C.; supervision, P.B.S.; project administration, P.B.S. All authors have read and agreed to the published version of the manuscript.

**Funding:** This research received no external funding.

**Data Availability Statement:** The data used cannot be shared neither publicly nor privately due to proprietary reasons.

**Conflicts of Interest:** The authors declare no conflict of interest.

## Abbreviations

The following abbreviations are used in this manuscript:

| | |
|---|---|
| EAF | Electric Arc Furnace |
| EE | Electrical Energy |
| VTD | Vacuum Tank Degasser |
| LRF | Ladle Refining Furnace |
| BOF | Basic Oxygen Furnace |
| CCM | Continuous Casting Machine |
| DRI | Direct Reduced Iron |
| ANN | Artificial Neural Network |
| SVM | Support Vector Machine |
| ML | Machine Learning |
| SOP | Standard Operating Procedure |
| RQ | Research Question |

**Nomenclature**

| | |
|---|---|
| $\bar{R}^2_{tr}$ | the mean adjusted coefficient of determination on training data for a model type |
| $\bar{R}^2_{te}$ | the mean adjusted coefficient of determination on test data for a model type |
| $R^2_{min,te}$ | the minimum adjusted coefficient of determination on test data for a model type |
| $R^2_{max,te}$ | the maximum adjusted coefficient of determination on test data for a model type |
| $Z_s$ | the set of 10 selected random state values |
| $N$ | the total number of data points in a dataset |
| $min_{tr}$ | the minimum number of training data points |
| $min_{te}$ | the minimum number of test data points |
| $S_n$ | number of sampled training and test datasets |
| $s$ | an arbitrary sample |
| $p^{tr}_{start}$ | the start point of a training dataset |
| $p^{te}_{start}$ | the start point of a test dataset |
| $p^{tr}_{end}$ | the end point of a training dataset |
| $p^{te}_{end}$ | the end point of a test dataset |
| $s_{tr}$ | the sample size of a training dataset |
| $s_{te}$ | the sample size of a test dataset |
| $r$ | Pearson correlation coefficient |
| $p$ | $p$-value of the Pearson correlation test |
| $M^{avg,j}_{200}$ | the 200-moving average curve |
| $n_j$ | the total number of stable models in interval $j$ |
| $N_v$ | the total number of 200-value intervals |

## Appendix A

**Table A1.** Variable groups for the LRF models.

| Variable Group | Variables | No. Variables |
|---|---|---|
| Base | First measured temperature<br>First predicted temperature<br>Last measured temperature<br>Last predicted temperature<br>Process time<br>Start weight<br>Addition weight<br>Ladle empty time<br>Electrode energy<br>Porous plugs volume<br>Lance stirring volume | 11 |
| Porous plugs time | Porous plugs time | 1 |
| Lance stirring time | Lance stirring time | 1 |
| Electrode time | Electrode time | 1 |
| Ladle trips | Ladle trips | 1 |

**Table A2.** Variable batches for the LRF models. Variable groups, as specified in Table A1, that included in a variable batch is marked with x. Empty cells indicates that the variable group is not included in the variable batch.

| Variable Batch | Base | Porous Plugs Time | Lance Stirring Time | Electrode Time | Ladle Trips |
|---|---|---|---|---|---|
| VB1 | x | x | | | |
| VB2 | x | x | x | | |
| VB3 | x | | | x | |
| VB4 | x | | | | x |
| VB5 | x | x | x | | |
| VB6 | x | x | | x | |
| VB7 | x | x | | | x |
| VB8 | x | | x | x | |
| VB9 | x | | x | | x |
| VB10 | x | | | x | x |
| VB11 | x | x | x | x | |
| VB12 | x | x | x | | x |
| VB13 | x | x | | x | x |
| VB14 | x | | x | x | x |
| VB15 | x | | | | |
| VB16 | x | x | x | x | x |

**Table A3.** Variable groups for the VTD models.

| Variable Group | Variables | No. Variables |
|---|---|---|
| Base | First measured temperature<br>First predicted temperature<br>Last measured temperature<br>Last predicted temperature<br>Process time<br>Start weight<br>Addition weight<br>Ladle empty time<br>Electrode energy<br>Porous plugs volume | 10 |
| Porous plugs time | Porous plugs time | 1 |
| Electrode time | Electrode time | 1 |
| Number of boosts | Number of boosts | 1 |
| Vacuum time | Vacuum time | 1 |
| Ladle trips | Ladle trips | 1 |

**Table A4.** Variable batches for the VTD models. Variable groups, as specified in Table A3, that included in a variable batch is marked with x. Empty cells indicates that the variable group is not included in the variable batch.

| Variable Batch | Base | Porous Plugs Time | Electrode Time | Number of Boosts | Vacuum Time | Ladle Trips |
|---|---|---|---|---|---|---|
| VB1 | x | x | | | | |
| VB2 | x | | x | | | |
| VB3 | x | | | x | | |
| VB4 | x | | | | x | |
| VB5 | x | | | | | x |
| VB6 | x | x | x | | | |
| VB7 | x | x | | x | | |
| VB8 | x | x | | | x | |
| VB9 | x | x | | | | x |
| VB10 | x | | x | x | | |
| VB11 | x | | x | | x | |
| VB12 | x | | x | | | x |
| VB13 | x | | | x | x | |
| VB14 | x | | | x | | x |
| VB15 | x | | | | x | x |
| VB16 | x | x | x | x | | |
| VB17 | x | x | x | | x | |
| VB18 | x | x | x | | | x |
| VB19 | x | x | | x | x | |
| VB20 | x | x | | x | | x |
| VB21 | x | x | | | x | x |

**Table A4.** *Cont.*

| Variable Batch | Base | Porous Plugs Time | Electrode Time | Number of Boosts | Vacuum Time | Ladle Trips |
|---|---|---|---|---|---|---|
| VB22 | x | | x | x | x | |
| VB23 | x | | x | x | | x |
| VB24 | x | | x | | x | x |
| VB25 | x | | | x | x | x |
| VB26 | x | x | x | x | x | |
| VB27 | x | x | x | x | | x |
| VB28 | x | x | x | | x | x |
| VB29 | x | x | | x | x | x |
| VB30 | x | | x | x | x | x |
| VB31 | x | | | | | |
| VB32 | x | x | x | x | x | x |

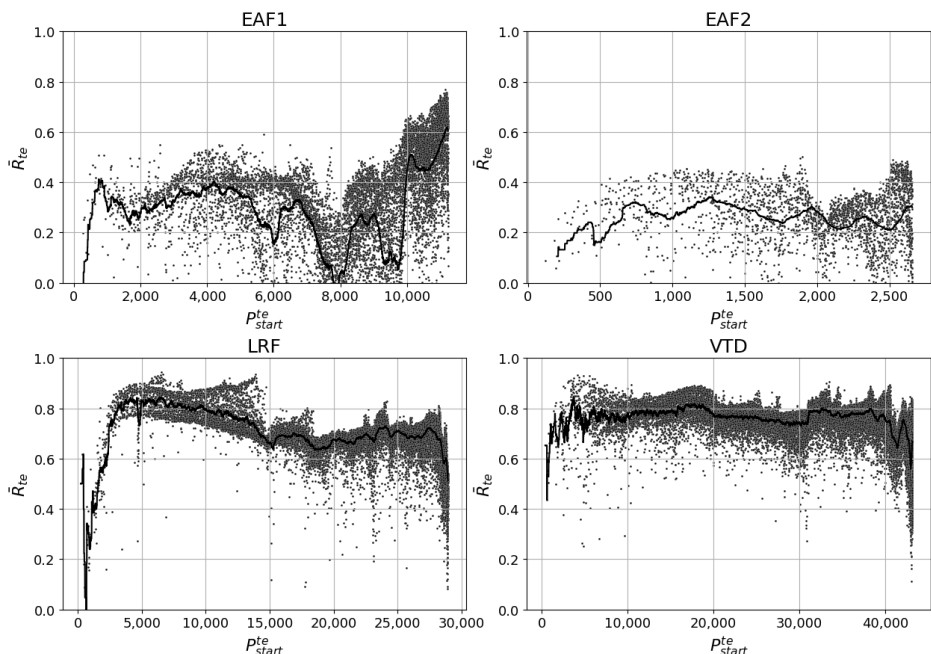

**Figure A1.** The mean R-squared on the test data ($\bar{R}^2_{te}$) plotted against the start of test set ($P^{te}_{start}$) for all models types created using the Unbiased Evaluation methodology.

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
