# Peer review of "A Proposed Methodology to Evaluate Machine Learning Models at Near-Upper-Bound Predictive Performance—Some Practical Cases from the Steel Industry"

_processes, doi:10.3390/pr11123447_

Round 1

Reviewer 1 Report

Comments and Suggestions for Authors

The present work aims to answer three essential research questions (RQ) that have previously not been explicitly dealt with in the field of applied machine learning (ML) in steel process engineering.

The three questions are:

How many training data points are needed to create a model with near upper-bound predictive performance on test data?

What is the near upper-bound predictive performance on test data?

For how long can a model be used before its predictive performance starts to decrease?

The ML has been widely used in steelmaking industries, however, the published work always show their ability to predict some variables or parameters with good agreement. The above mentioned three questions are very important for ML and steel industry.

The manuscript is well organized, fits the scope of the journal Processes and the topic is very important. The quality of the conducted study is high. The presented results of the performed modeling work have scientific and practical meaning. A literary review has been made quite fully. The description of models, research background are solid. A demonstration of practical application and motivation in the introduction would greatly improve the article.

I recommend to accept this manuscript as the present form. The following comments may be useful for a future improvement of this study.

1.       The data sets from both LRF and VTD represents production from year 2008 until the middle of year 2022. This is very good to track so many data sets. Would the steel ladle capacity expands or many other issues happened during this years affect the results?

2.       The predictive performance metric used in the present work is the adjusted R square. Could some else parameter be used for evaluation purpose?

3.       From table 2, most parameter in steelmaking process was used in the model. Since LF and VD operation are steel grade based, and the temperature, plug volume (stirring intensity), vacuum time are closely related to steel grades. The authors mentioned the steel grades are wear plates, what is the common composition of steel? Would the authors think about sub group the data with reference to steel grade? However, this is not mandatory and the authors may think about this later on.

4.       Maybe this manuscript is too long for readers.

Author Response

Dear reviewer,

Please view the attached file for our response.

Reviewer 2 Report

Comments and Suggestions for Authors

This manuscript introduces a methodology addressing prevalent research questions in applied machine learning within steel process engineering. I have the following comments that the authors should carefully address in the revision.

1. Enhance Literature Review: The introduction discussed some prior works employing machine learning models in the steel industry. However, it is crucial to elucidate the rationale behind employing machine learning models within this specific sector. Clarifying why machine learning is pertinent in the steel industry would augment the manuscript's foundation.

2. Simplify Upper-Bound Predictive Performance Definition: The definition presented in Section 2.1 outlining the upper-bound predictive performance is comprehensive but appears overly intricate.

3. Optimize Parameter Search: The extensive grid search for optimal parameters, as evidenced by the 1536 and 3072 model types for LRF and VTD in Section 3.2.4 can be resource-intensive and time-consuming. It is recommended to explore automated hyperparameter optimization frameworks like Optuna to simplify this process. 

Author Response

(The authors gave the same response as above.)

Reviewer 3 Report

Comments and Suggestions for Authors

I think that before publishing this paper a number of issues should be clarified:

1) When conducting a methodological study, it is crucial to provide a thorough explanation of the reproducibility and transferability issues associated with the proposed approaches for future data analysis. Nonetheless, the Data Availability Statement indicates that the data utilized cannot be shared, either publicly or privately, due to proprietary reasons. To enhance the paper's quality and achieve transparency in the research process, could you please provide a non-proprietary example of how you implement your methodology using model data? It would be convenient for me to review a csv file of the dataset, an ipynb file with the source code, and a step-by-step guide of your proposed methods. Please be precise in your specification of imported package versions, if applicable.

2) Regarding your response to the first research question, it was stated that the percentage of training data required to produce a model with near upper-bound prediction accuracy on test data differs greatly across the four data sets, ranging from 15.5% to 92.7%. This significant disparity is worth highlighting. At the same time, the average R-squared on the test data ($\bar{R^{2}_{te}}$) plotted against the number of data points in the training set does not provide a clear indication of the optimal number of data points for training. Can you suggest any modifications to the metric to increase its sensitivity to deviations from the optimum? Additionally, it would be beneficial to display the Training Loss vs Validation Loss curves for the chosen optimal models.

3) The study employed logistic and tanh activation functions. Additionally, ReLU, one of the most commonly used activation functions, was assessed due to its significant benefits in detecting potential nonlinear responses on hidden layers of the neural network. Was ReLU taken into account during the analysis? If so, what findings were observed? If not, what were the reasons for its exclusion? I would appreciate further information regarding the significance of normalizing data prior to training, selecting the most appropriate hyperparameters, preventing overtraining and implementing regularization techniques.

Author Response

(The authors gave the same response as above.)

Round 2

Reviewer 3 Report

Comments and Suggestions for Authors

First, it is great that the authors strive for transparency and
improve the quality of the paper by providing the code used in the
experiments as supplementary information. This not only improves the
quality of the paper, but also provides transparency of the research
process, promoting reproducibility.

The authors have pointed to Table 6 as well as the section "5.
Conclusions" where significant differences in the percentage of
training data are discussed. We appreciate the authors' effort to
answer this question and provide references to support these
observations.

It is good to see that the authors were responsive to our suggestions,
especially in their explanation of the rationale for using three
metrics to determine the number of training data points needed for
prediction. Their willingness to consider modifications to the metric
to increase its sensitivity to deviations from the optimum reflects a
forward-thinking approach.

In the context of evaluating activation functions, the authors'
decision to limit the grid search to tanh and logistic functions to
ensure consistency across the four datasets was justified in these
responses. The rationale for rejecting ReLU combined with
consideration of its inclusion and evaluation in a grid search format
is also valid.

The authors' detailed explanation of the steps taken to select
appropriate hyperparameters, mitigate overtraining, and use
standardization to normalize variables is accepted.

In conclusion, the authors' responses demonstrate a commitment to
transparency, meticulousness in the study methodology, and an interest
in possible future improvements. Their work reflects a deep
understanding of the issues at hand, and the thoughtfulness of next
steps increases the potential impact of their study.